# Supervised SVM Transfer Learning for Modality-Specific Artefact Detection in ECG

**DOI:** 10.3390/s21020662

**Published:** 2021-01-19

**Authors:** Jonathan Moeyersons, John Morales, Amalia Villa, Ivan Castro, Dries Testelmans, Bertien Buyse, Chris Van Hoof, Rik Willems, Sabine Van Huffel, Carolina Varon

**Affiliations:** 1STADIUS Center for Dynamical Systems, Signal Processing and Data Analytics, Department of Electrical Engineering (ESAT), KU Leuven, 3001 Leuven, Belgium; johnfredy.moralestellez@esat.kuleuven.be (J.M.); amalia.villagomez@kuleuven.be (A.V.); sabine.vanhuffel@kuleuven.be (S.V.H.); carolina.varon@kuleuven.be (C.V.); 2IMEC, 3001 Leuven, Belgium; ivand.castro@imec.be (I.C.); chris.vanhoof@imec.be (C.V.H.); 3Department of Chronic Diseases, Metabolism and Ageing, KU Leuven, 3001 Leuven, Belgium; dries.testelmans@uzleuven.be (D.T.); bertien.buyse@uzleuven.be (B.B.); 4Department of Cardiovascular Sciences, University Hospitals of Leuven, 3001 Leuven, Belgium; rik.willems@uzleuven.be; 5e-Media Research Lab, Department of Electrical Engineering, KU Leuven, 3001 Leuven, Belgium

**Keywords:** transfer learning, artefact detection, signal quality, ECG analysis

## Abstract

The electrocardiogram (ECG) is an important diagnostic tool for identifying cardiac problems. Nowadays, new ways to record ECG signals outside of the hospital are being investigated. A promising technique is capacitively coupled ECG (ccECG), which allows ECG signals to be recorded through insulating materials. However, as the ECG is no longer recorded in a controlled environment, this inevitably implies the presence of more artefacts. Artefact detection algorithms are used to detect and remove these. Typically, the training of a new algorithm requires a lot of ground truth data, which is costly to obtain. As many labelled contact ECG datasets exist, we could avoid the use of labelling new ccECG signals by making use of previous knowledge. Transfer learning can be used for this purpose. Here, we applied transfer learning to optimise the performance of an artefact detection model, trained on contact ECG, towards ccECG. We used ECG recordings from three different datasets, recorded with three recording devices. We showed that the accuracy of a contact-ECG classifier improved between 5 and 8% by means of transfer learning when tested on a ccECG dataset. Furthermore, we showed that only 20 segments of the ccECG dataset are sufficient to significantly increase the accuracy.

## 1. Introduction

An electrocardiogram (ECG) measures the electrical activity of the heart. It is used by cardiologists to obtain a better understanding of the heart’s functioning. A typical cardiac examination is taken in a hospital environment and lasts only 10 s. This is often sufficient to detect major pathology’s, yet this “snapshot” can be deceptive when used to evaluate one’s general condition [1].

A solution for this problem is to monitor the patient outside of the hospital, during a longer period of time [2]. Due to the extension of the analysis period, the detection rate of cardiac events can be highly increased, compared to the “snapshot” ECG in the hospital. However, recording vital signals outside of the hospital environment increases the likelihood of the signals being exposed to noise. During a cardiac examination in the hospital, the patient is asked to lay still in a supine position, and thus, as a result, the recorded signals are generally of very high quality. This is no longer the case for ambulatory recordings. The diagnostic capabilities of the signals can be reduced by the presence of artefacts in the signal. These contaminants can be caused by electrode motion, contact noise and electromyography among others [3].

One of the most commonly used wearable devices is a Holter. It continuously records the electrical activity of the heart for a period of, typically, 24 or 48 h. Although typical electrodes are used, the device can be worn around the hip. This allows the patient to participate in daily life activities, in contrast to the “snapshot” ECG in the hospital. Although the extension of the analysis period highly increases the detection rate of specific cardiac events, it is still not enough to detect a large amount of cardiac conditions [1,4]. It is stated in [5] that recordings acquired during up to 3 months are beneficial for this task.

Capacitively coupled ECG (ccECG) sensing is an unobtrusive monitoring technique which can be used to extend the analysis period further. It enables ECG signals to be recorded through insulating materials, such as clothing, without direct skin contact. This allows the capacitive electrodes to be integrated into daily life objects, such as mattresses or (car) seats [6,7]. On the upside, this provides more comfort for the patient. However, on the downside, this introduces new, modality-specific, artefacts originating from electromagnetic changes in the surroundings, movement of the patient and poor electrode coupling. These artefacts can be different from those in contact ECG [8]. Therefore, it might be that an artefact detection model, trained on contact ECG, cannot accurately distinguish clean from noisy ccECG signals without any adaptation.

One way to solve this, is to train and test a classifier on labelled ccECG data. Multiple artefact detection algorithms have been developed in recent years. Examples of the used methodologies are QRS amplitude ratios in combination with Principal Component Analysis (PCA) [9], wavelet decompositions [10] and frequency features in combination with Independent Component Analysis (ICA) [11]. Moreover, in [12], the authors derived basic features from pressure and ccECG signals to accurately separate clean from noisy segments.

Castro et al. obtained an accurate separation by using only signal quality features for contact ECG, such as kurtosis and spectral density ratio’s [8]. A downside of this approach is the requirement of a lot of ground truth data. This is usually obtained by manually labelling the signals, which is a tedious and time consuming task. A better and faster solution would be to train a classifier on contact ECG data and adapt it for ccECG data. As many labelled contact ECG datasets exist, this presents a more feasible solution.

We propose to use transfer learning to extend a model from contact ECG to other sensor modalities, like ccECG, without the need of relabelling an entire new dataset. This technique exploits the knowledge obtained from a previous task to optimise the performance on a similar target task, for which fewer labelled data points are available [13]. It thereby circumvents the need to retrain a new classifier from scratch and requires only a limited amount of labelled data. Applied to our case, it means that an artefact detection model for one signal modality, namely, contact ECG, can be adapted with transfer learning towards another signal modality, ccECG in this case. Of course, this would only be needed if an artefact detection model, trained on contact ECG, performs poorly on a ccECG dataset.

The first goal of this study was to investigate the necessity of transfer learning. We used features that were previously used for artefact detection in contact ECG [14], to construct an artefact detection model. Then, we investigated its ability to make an accurate distinction between clean and noisy ccECG samples. The second goal was to optimise the performance of this artefact detection model towards ccECG. To this end, we made use of the transfer learning formulation described in [15], to create a modality-specific classifier by using only a limited amount of modality-specific data. Last, we investigated whether an active sampling strategy based on entropy could decrease the amount of samples that are needed to optimise the model.

To the best of our knowledge, this is the first study that showed that an artefact detection model for one signal modality, namely, contact ECG, can be optimised with transfer learning towards another signal modality, ccECG in this case.

## 2. Materials and Methods

### 2.1. Datasets

We used ECG recordings from three different sources, acquired with three different recording devices: (1) a polysomnographic dataset, recorded in the University hospitals Leuven (UZ Leuven), Belgium; (2) a handheld dataset, available in Physionet and recorded for the Computing in Cardiology (CinC) Challenge of 2017; and (3) a capacitively coupled dataset, which originated from multiple sources. The first two datasets consist of contact ECG, albeit recorded with a different device, and the third consists of ccECG. In the following paragraphs, we will outline the datasets. A complete overview is given in Table 1.

#### 2.1.1. Polysomnographic Dataset (PSG)

This dataset was recorded in the sleep laboratory of the University hospitals Leuven, UZ Leuven, Belgium. The study protocol was approved by the Ethical Committee of the University Hospitals of Leuven (S53746). It consists of 16 single-lead (lead II) ECG recordings from 16 patients, accounting for a total of 152 h and 12 min of data. A sampling frequency of 200 Hz was used.

A medical doctor evaluated each minute for the presence of artefacts and assigned a binary label (+1 = clean, −1 = noisy). This procedure resulted in a total of 9132 one-minute signals of which 295, or 3.2%, are noisy. These labels have been previously used for artefact detection in [8,16].

#### 2.1.2. Handheld Dataset (HH)

The PhysioNet/Computing in Cardiology (CinC) Challenge of 2017 was intended for the differentiation of atrial fibrillation (AF) from noise, normal and other rhythms in short term ECG recordings [17]. All signals were recorded by volunteers with an AliveCor’s single-channel ECG device. They were instructed to hold one electrode in each hand, thereby creating a lead I (LA-RA) equivalent ECG. The signals lasted between 9 and 60 s, were sampled at 300 Hz, and stored as 16-bit files with a bandwidth of 0.5–40 Hz.

In total, the dataset consists of 12,186 ECG signals, of which 8528 were provided for training. Only the normal rhythm and noisy class from the training set were of interest for this paper. They accounted for a total of 5334 signals. A more detailed description of the dataset can be found in [17].

The organisers of the CinC challenge indicated that the labels of the signals could be improved. Therefore, in [14] a relabelling procedure was implemented. Each signal was labelled by four independent annotators according to the following rule.

“If the annotator is able to confidently distinguish all R-peaks in the signal, then the signal is labelled as clean. Otherwise, it is labelled as contaminated.”

Only the signals on which all annotators agreed, were considered for further analysis. They accounted for a total of 3857 signals, of which 62.3% are clean.

#### 2.1.3. Capacitively Coupled Dataset (CC)

This dataset consists of 10,000 ccECG signals originating from two recording devices. The first recording device is described in [18,19] and the second one was used to create the publicly available UnoVis dataset [7]. The dataset comprises ccECG signals from different scenarios, such as signals recorded while sitting in a static car seat, while driving a car, while sleeping and while working on an office chair. Each signal was 15 s long with a sampling frequency of 512 Hz.

All signals were labelled by five annotators with experience in ECG signal analysis. Three classes were used: (1) Useless or no ccECG; (2) ccECG with artefacts that may affect the detection of two to five heartbeats; (3) ccECG useful for heart rate variability (HRV) analysis and possibly morphology analysis.

In total, 90 signals with strong annotation disagreement, labelled as both 1 and 3 by different annotators, were discarded. This resulted in a dataset of 9910 signals on which at least three annotators agreed. A Fleiss’ Kappa of 0.80 was obtained.

The three quality levels were assigned to two binary classification problems: one with a “low threshold” (level 1 vs. levels 2–3) and another with a “high threshold” (levels 1–2 vs. level 3). The high threshold labelling matches the best with the labelling procedure in the other studies. Therefore, we conducted all the analysis with these labels. The selected labelling procedure resulted in 80.5% noisy signals and 19.5% clean signals.

### 2.2. Preprocessing

We filtered all signals with a zero phase 2nd-order high- and 4th-order low-pass Butterworth filter with cut-off frequencies at 1 and 40 Hz, respectively. This ensures the removal of baseline wander and high frequency noise, and retains the relevant physiological information.

### 2.3. Features

We used the same feature crafting methodology as in [14]. As a first step, we moved a sliding window over the ECG signals with a width of five seconds and 80% overlap. As a second step, we computed the autocorrelation function (ACF) from each sliding window. This was obtained by computing the inverse Fourier transform of its corresponding power spectral density (PSD) [21]. We used a maximum time lag of 250 ms to exclude consecutive heartbeats and to ensure the inclusion of all characteristic waveforms.

The most characteristic waveform of an ECG signal is the QRS-complex. It consists of three consecutive deflections: a downward deflection, the Q-wave; an upward deflection, the R-peak; and another downward deflection, the S-wave. As we use a maximum time lag of 250 ms, we can assume that the shape of the ACF is primarily defined by a shift of this complex. An example of the ACF of two contact and non-contact signals with a different quality level can be seen in Figure 1. From the ACF we can derive three intuitive features.

First (local) minimum (FMin): As the QRS-complex consists of three opposite deflections, we can assume that by shifting the QRS-complex, we reach a time lag where the opposite deflections overlap. Due to the high amplitude of the QRS-complex, this shift should coincide with the first local minimum in the ACF.A large shift, could be an indication of a flat line, while a small shift could be an indication of a sharp, high amplitude, artefact. In Figure 2, a large, wide artefact is depicted. This results in a large FMin, which indicates that the lag is a lot larger than is normally expected.To derive a feature for the entire segment, we first selected the location of every first local minimum for all sliding windows. Afterwards, the overall minimum of the whole segment was computed. This results in a single value per segment.Maximum amplitude at 35 ms (MAmp): For the previous feature, we look at the shift that results in a local minimum. We do not make an assumption as to where this local minimum should be. Furthermore, we do not look at the amplitude of the ACF at this time lag. For the second feature, we make the assumption that the first local minimum of an average clean segment is around 35 ms. This time lag was empirically defined in [14]. We take the amplitude of the ACF at this time lag as second feature.High amplitudes, as well as large time lags of the previous feature, could indicate a flat line. They could also indicate a technical artefact with a high amplitude and large width (Figure 2).To have a general feature for the entire segment, we select the amplitude at 35 ms in the ACF of all sliding windows and we take the maximum over all obtained values.Similarity (Sim): An ECG signal is a very repetitive, quasi-stationary, signal. While the heart rate may differ along the signal, the morphology of the heartbeat remains comparable. Therefore, it is safe to assume that a measure of similarity would be a good indication of the quality of an ECG signal.From the ACF’s of all sliding windows, we selected the interval between time lags 30 and 115 ms. The boundaries of these intervals were empirically defined. The amplitudes before 30 ms could be of interest, but are already taken into account in the first two features. Additionally, we observed no added value in terms of classification performance by extending the interval beyond 115 ms. Hereafter, we computed the pairwise euclidean distance between all the ACF’s in this interval and used the maximum of these distances as a measure of similarity. We can observe the difference between a clean and a noisy non-contact ECG signal in Figure 3.A large value of this feature indicates the presence of a divergent ACF. It could also be an indication of variable ECG morphologies within that segment. In essence, the larger the value of this feature, the less similar the ACF’s in that interval are and the more likely that the quasi-stationarity, that is expected in an ECG signal, is destroyed by the presence of an artefact.

### 2.4. Classification

In this section, we describe the base classifier and the theory behind the optimisation of that classifier using transfer learning. Additionally, we explain the active sampling strategy based on entropy that we want to investigate.

#### 2.4.1. Base Classifier

Support vector machines (SVM) classification models with an RBF Kernel were used as base classifiers for the transfer learning. These classifiers are adapted using the transfer learning formulation described in [15].

Consider a training set ${\left\{x_{i},y_{i}\right\}}_{i=1}^{N}$, with $x_{i}\in R^{3}$, as the input data, and $y_{i}\in \left\{-1,+1\right\}$ as the corresponding labels. In this case, class +1 ($y_{i}$ = 1) represents clean samples and class −1 ($y_{i}$ = −1) noisy samples. The SVM maps the input data to a high dimensional feature space using a mapping function ϕ(x). This transformation allows to separate the datapoints in a higher dimensional space using the hyperplane $w^{T}\phi \left(x\right)+b$, with *w* the unknown weight vector and *b* an unknown bias term.

The SVM can be formulated in the primal space as
(1)$$\begin{array}{cc} \; & \min_{w,b,\xi }J\left(w,\xi \right)=\frac{1}{2}{\left||w|\right|}^{2}+C\sum_{i=1}^{N}\xi_{i}\\ s.t. & \left\{\begin{array}{c} y_{i}\left(w^{T}\phi \left(x_{i}\right)+b\right)\geq 1-\xi_{i}\\ \xi_{i}\geq 0\;,\forall i\in \left[1,N\right], \end{array}\right. \end{array}$$
with *w* as the weight vector of the base classifier and $\xi_{i}$ as the classification error of the model on data point $x_{i}$. The regularisation constant *C* provides a balance between regularisation and minimisation of the number of misclassifications: when *C* is too high, this will lead to overfitting, and when *C* is too low, this causes underfitting. In this paper, the hyperparameters are tuned using Bayesian optimisation in combination with fivefold cross-validation [22].

As the datasets are rather small, we opted to use five fold cross-validation to evaluate the classifiers. The average result was used as a reference for the transfer learning classifier.

Additionally, as we are also interested in the inter dataset performance, we evaluated the classification model of each fold of one dataset, on all folds of the other datasets (Figure 4). This results in 25 performance evaluations (5×5 folds). It provides an indication of the similarity of the datasets and the overall performance of the classification model.

#### 2.4.2. Transfer Learning

We optimised each base classifier for the other datasets by applying the transfer learning approach described in [15]. The key concept of this approach is the modification of the objective function of the SVM. This is constructed so that it minimises both the classification error on the new training data and the discrepancy between the adapted and base classifier.

Let ${\left\{\tilde{x_{k}},\tilde{y_{k}}\right\}}_{k=1}^{M}$ be the *M* training points of the new dataset, with $\tilde{x_{k}}\in R^{3}$ as the input data, and $\tilde{y_{k}}\in \left\{-1,+1\right\}$, the corresponding labels. De Cooman et al. proposed the following optimisation problem for an adapted SVM classifier [15],
(2)$$\begin{array}{cc} \; & \min_{\tilde{w},\tilde{b},\tilde{\xi }}J\left(w,\xi \right)=\frac{1}{2}\left|\right|\tilde{w}-w{\left|\right|}^{2}+D\sum_{k=1}^{M}\tilde{c_{k}}\tilde{\xi_{k}}\\ s.t. & \left\{\begin{array}{c} \tilde{y_{k}}\left({\tilde{w}}^{T}\phi \left(\tilde{x_{k}}\right)+\tilde{b}\right)\geq 1-\tilde{\xi_{k}}\\ \tilde{\xi_{k}}\geq 0\;,\forall k\in \left[1,M\right], \end{array}\right. \end{array}$$
with $\tilde{w}$ the weight vector of the adapted classifier, *D* the regularisation constant, similarly to *C* in Equation (Equation 1), $\tilde{c_{k}}$ an additional weight constant and $\tilde{\xi_{k}}$ the error of the model on data point $\tilde{x_{k}}$. The factor $\tilde{c_{k}}$ was used in [15] to counter the class imbalance in the dataset. In this study, we always selected an equal amount of samples from both classes, so that the subsets for transfer learning were always balanced. This results in $\tilde{c_{k}}=1,\forall k\in \left[1,M\right]$.

Hyperparameter *D* allows to balance between minimising the errors for the new training data points and minimising the discrepancy with the base classifier. De Cooman et al. empirically adjusted *D* based on the amount of new data points, for example, if more than 10 new datapoints were available, *D* was set to 100 [23]. Therefore, as we always have more than 10 new datapoints available, we also used a value of 100. This value might still be fine tuned, but an in-depth analysis of the optimal *D* is outside the scope of this paper.

The advantage of transfer learning is that it allows to adapt an existing classifier with a limited amount of data [13]. Therefore, instead of using the whole training set ${\left\{\tilde{x_{k}},\tilde{y_{k}}\right\}}_{k=1}^{M}$, we can use a subset ${\left\{\tilde{x_{l}},\tilde{y_{l}}\right\}}_{l=1}^{P}$, with P<<M. Ideally, this subset represents the main characteristics of the whole training set. In [15], a random subset selection was used. Therefore, to objectively evaluate the results on the given problem, we applied the same sampling procedure here.

We used 500 randomly selected samples—250 clean and 250 noisy—of the training folds of the CC dataset to update the base classifiers of the other two datasets. These were selected 10 times for generalisation of the results.

#### 2.4.3. Subset Selection

The amount of training samples available for transfer learning can significantly affect the performance of the algorithm. Therefore, this subset should be a fair representation of the underlying distribution of the whole dataset. With this in mind, we investigated the added value of an active sampling strategy, called fixed size sampling, which is based on quadratic Rényi entropy ($H_{r}$) [24]. This is defined as
(3)$$H_{r}\left(x\right)=-log\int p{\left(x\right)}^{2}dx$$
with *p* as the probability density function. This can be estimated as
(4)$$\int \hat{p}{\left(x\right)}^{2}dx=\frac{1}{N^{2}}1_{N}^{T}\Omega 1_{N},$$
with *N* as the number of samples, Ω as the kernel matrix and vector $1_{N}$ as a Nx1 dimensional vector [25].

This sampling strategy creates a subset with an entropy that approximates the entropy of the entire training set. It consists of the following steps.

First, a subset with size P(P<<M) is randomly selected. Second, a random sample of the subset is swapped with a new, randomly selected, sample of the training set. If the quadratic Rényi entropy of the subset increases, then the new sample replaces the old sample in the subset. Otherwise, the exchange is rejected. These steps are repeated until there is no more increase in entropy or after a predefined number of iterations, which is set to 1000. This results in an entropy value of the subset that approximates the entropy of the entire training set.

We compared the random sampling and fixed size sampling for a varying number of subset sizes. More specifically, we used a subset of 20, 50, 100, 200 and 500 randomly selected samples from the new training set to adapt the classifier. The classes were equally divided over each set. We hypothesise that the fixed size sampling approach results in a steeper learning curve, compared to the random sampling approach. In order to make a general conclusion, we repeated this procedure 10 times for each subset size. In order to make a general conclusion, we repeated this procedure 10 times for each subset size. This results in a total of 250 performance evaluations for each subset size. Namely, 25 folds, similar to the performance evaluation in Section 2.4.1, times 10 repetitions per subset size.

### 2.5. Performance Evaluation

We assessed the performance of the algorithms using the accuracy, (Acc, percentage of correctly detected samples), sensitivity (Se, percentage of correctly detected clean samples) and specificity (Sp, percentage of correctly detected noisy samples). Additionally, as the class distribution of the datasets is unbalanced, we used the balanced accuracy (bAcc) to compare the performances [26]. This is computed by taking the average of the Se and Sp. We used paired *t*-tests to compare the two sampling methodologies. A p<0.05 was considered significant.

## 3. Results

The feature spaces of the three datasets are depicted in Figure 5. It can be observed that the feature space of the PSG is more compact compared to the other two. Additionally, the three plots also show a shift in the data distribution. The PSG contains mostly clean samples (96.8%), while the HH contains only 62.3% clean samples, and the CC contains mostly noisy samples (80.5%). The latter is due to the inclusion of “floating measurements”, in the dataset. These are measurements in which the device was turned on, but without a user present [8]. These two observations pose an extra difficulty for the base classifier, as it assumes the same class distribution in the test set, with respect to the training set.

Table 2 gives an overview of the results of the base classifiers. We obtained very good results for both contact and capacitive ECG databases (bAcc: 87.5–90.3), when the training and testing subsets are part of the same dataset. However, these similar bAcc’s mask the underlying distribution of the Se and Sp values. The balance between Se and Sp changes drastically over the three datasets. We can observe high Se values for the PSG dataset and high Sp values for the CC dataset. Similarly, the CC model obtains a higher Sp compared to Se when tested on the other datasets.

Figure 6 shows the average performance of the base classifiers on the CC dataset compared to the transfer learning approach. For the latter, 500 (250 clean and 250 noisy) randomly selected samples were used to update the base classifiers and the whole procedure was repeated 10 times. These results show that the transfer learning classifiers outperform the base classifiers for almost all performance metrics. Only the Se of the base classifier, trained on the HH dataset, decreased. The strongest increase can be observed for the Sp.

As a next step, we investigated the influence of the number of included training samples. Figure 7 shows the effect of the number of samples included for training (x-axis) on the different performance metrics (y-axis). We only show the performances up to a training set of 200 samples, since no improvement was obtained hereafter. Despite a clear difference in entropy (Figure 8), the only significant improvements of the fixed-size sampling on the PSG base classifiers could be observed in the Se when 100 and 200 samples were included. Moreover, significantly lower Acc and Sp values were obtained with a subset of only 20 samples. No significant differences were obtained for the HH base classifiers.

The Se of the PSG base classifier decreased drastically when 20 training samples were included. From this value, the sensitivity of the PSG base classifier increased significantly until the training subset contained 200 samples. No significant changes occurred hereafter. The Sp increased significantly up to 50 samples, and seized to increase hereafter. The resulting bAcc followed the same V-shaped pattern as the Se. After an initial drop at 20 training samples, it increased significantly for every addition of extra samples. Starting from a subset of 100 samples, the median bAcc of the transfer learning classifier was higher than the base classifier. Additionally, the variability of all performance metrics decreased for every addition of extra samples to the subset. This can be observed by the narrowing of the interquartile ranges.

We observed a similar pattern for the base classifier trained on the HH dataset. For example, the Se also showed a V-shaped pattern, albeit not so outspoken as the other base classifier. When more samples were added to the subset, the Se increased significantly up to a subset of 100 samples. However, it never reaches the same Se as the base classifier. A significant increase in Sp could only be observed between a subset of 50 and 500 samples. The balanced accuracy increased with every addition of more samples, except between 100 and 200 samples. An increase could be observed between these two values, but it was not significant. In contrast to the other base classifier, here the median bAcc of the transfer learning classifier was already higher after the inclusion of 20 samples in the training subset.

As an extra experiment, we wanted to see how much the transfer learning approach could improve the lowest performance metric from Table 2. The lowest value was obtained for the Se of the CC base classifier, when tested on the HH dataset. Therefore, we applied the transfer learning approach on the CC classifier and incrementally added more training samples from the HH dataset. A strong increase could already be observed after including only 20 samples (Figure 9). This increase continued with the inclusion of more samples, together with a decrease in Se variability.

## 4. Discussion

### 4.1. Features and Base Classifier

The primary goal of this study was to optimise the performance of a simple artefact detection model, that is trained on contact ECG, towards ccECG. To this end, we used three features that have previously shown to produce good results for artefact detection in contact ECG [14].

In order for a transfer learning approach to be successful, we need to be sure that the proposed features are able to make a distinction between clean and noisy samples. Visually, this seems to be the case for the PSG dataset, as little to no overlap between the two classes can be observed in the feature space (Figure 5). However, this overlap increases for the HH dataset and even more for the CC dataset. Quantitatively, the little overlap of the PSG dataset results in a high performance of the base classifier on the test set of the PSG dataset (Table 2). We obtained a bAcc of 90.27±6.0. Remarkably, despite the visible overlap in the feature space, a similar bAcc was obtained for the two other datasets. This indicates that the proposed features are able to make an accurate distinction between clean and noisy samples, regardless of the recording modality. These results confirm the findings in [14], where accurate results were obtained when a classifier was trained on a subset of the PSG dataset and tested on an independent subset of the HH dataset. However, the results in this study allow for a more trustworthy conclusion, due to the robustness of the analysis.

The bAcc of the CC base classifier, when tested on the CC testing folds is on average 87.5±1.8 (Se = 80.0±2.7, Sp = 94.9±1.2). Despite this high accuracy, it is lower compared to the results obtained on the same dataset in [8]. There, a bAcc of 94.02 (Se = 95.19, Sp = 92.85) with a linear SVM classifier was obtained. In that paper, only three features, which were selected with a threshold-based one-level decision tree (DT) approach, were used. However, one important difference exists between these studies. Castro et al. used only one random training and testing split. Therefore, it is hard to draw general conclusions from these results. In this study we performed a 5-fold cross-validation, which produces results that should generalise better.

In the introduction, we hypothesised that an artefact detection model, trained on contact ECG, and without any adaptation, cannot accurately distinguish clean from noisy ccECG signals. We assumed that this would be due to the artefact characteristics of the different recording modalities [8]. The resulting Acc values seem to confirm this hypothesis, as they are remarkably lower, compared to the values of the original classifier. This is due to a substantial decrease in Sp, sa the Se values increased. These findings indicate that the clean ccECG segments are accurately detected, but that the noisy segments are not. Therefore, one could assume that this confirms the initial hypothesis and is indeed caused by a difference in artefact characteristics. However, that assumption would ignore another important factor, namely, the difference in class distribution between the different datasets. For example, the distribution of the PSG dataset is drastically different from the CC dataset. The PSG dataset contains predominantly clean segments (96.8%), while the opposite is true for the CC dataset (19.5%). This information does not exclude the difference in artefact characteristics as an influencing performance factor, but highlights the fact that the class distribution is also a major performance influencing factor.

### 4.2. Transfer Learning

The second goal of this study was to optimise the performance of the artefact detection model towards ccECG. The used transfer learning approach has previously shown its added value for the two previously stated performance influencing factors. For instance, in [15], it was used to adapt a patient-independent seizure detection algorithm towards a patient-dependent solution. De Cooman et al. have shown that the inclusion of a limited number of patient-specific seizures can result in a strong increase of performance for most patients [15]. In other words, a patient-independent model, which performance is otherwise too low for practical use, was successfully optimised with the proposed transfer learning approach, using patient-specific seizure characteristics.

The approach that they used was an adaptation of the transfer learning approach developed in [27]. Originally, it was intended to adapt existing classifiers to a new set of data with a different distribution, much like the problem in this study. They have shown that their method outperforms several standard methodologies, when it is applied on an artificial and a benchmark classification task. Therefore, regardless of the reason why the base classifiers fail to accurately detect the noisy segments of the CC dataset, the transfer learning approach should be able to improve the classification performance.

From Figure 6, we can observe that this is indeed the case. The performance of the PSG and HH base classifiers on the CC dataset significantly increased when transfer learning was applied. This indicates that the transfer learning approach successfully improved the performance of a classifier trained on contact ECG. The only performance metric that failed to increase was the Se of the HH dataset. This could be explained by the very high Se value for the base classifier. Visually, this means that the decision boundary of the HH base classifier is directed more towards the centroid of the noisy class in the CC dataset (Figure 5). This results in a very high Se and a low Sp. However, when transfer learning is used, this decision boundary presumably shifts towards the middle between the two classes. This shift results in a higher Sp and a lower Se.

In this work, we used a simple approach with only three features to show the relevance of the transfer learning. Future work should consider the addition of other features in order to improve the performance even further.

#### Subset Selection

We have shown that transfer learning allows to train a device-specific classifier, while using only a limited amount of device-specific data. More specifically, we used 500 labelled samples to obtain the results depicted in Figure 6. As labelling is a very tedious and time-consuming task, it is important to know the minimum amount of samples that are necessary to have an acceptable accuracy. To this end, we performed an experiment in which we varied the amount of randomly selected training samples from 20 to 200. Additionally, we performed the same experiment, using an active sampling approach, based on entropy. We hypothesised that the active sampling approach would result in a steeper learning curve, compared to the random sampling approach.

Remarkably, the only significant improvements of the PSG base classifiers by the active sampling could be observed in the Se when 100 and 200 samples were included. No significant differences could be observed for the other sampling sizes and performance metrics. On the contrary, significantly lower Acc and Sp values were obtained with a subset of only 20 samples. Therefore, despite a higher entropy for all training subset sizes, the proposed active sampling strategy did not have a steeper learning curve and did not reach the performance plateau faster.

This could be because the samples at the edges of the feature space are more likely to be selected since they increase the Rényi quadratic entropy of the subset. However, these samples might poorly represent the underlying distribution of the whole dataset. Tong et al. have shown that selecting samples close to the decision boundary can improve classification performance [28]. Therefore, in future research, we should investigate such approaches.

From the current analysis, we can state that an increase in entropy does not result in an increase of informative samples. Moreover, we can conclude that active sampling based on entropy does not provide an added value for transfer learning, compared to random sampling.

In Figure 7, we can observe similar patterns for the Se and Sp for the two base classifiers. Namely, a V-shaped pattern for the Se and a mirrored version for the Sp. This pattern can be explained by an initial overfitting, when a small training subset is used, and an incremental increase in performance when more training samples are included. However, this increase seems to stagnate when more training samples are included. In other words, the performance reaches a plateau. For both base classifiers, this appears to be when the training subset contains 200 samples (100 samples of each class).

### 4.3. Limitations

A limitation of this study is the composition of the CC dataset. We used data from two ccECG recording devices, which recorded signals in various scenarios, including floating sensors. We made the assumption that the difference in artefact characteristics can be related to the distinction between contact and non-contact ECG. However, it might be that the recording devices and scenario’s in which they were used also influence these characteristics. In a future study, this should be investigated into more detail.

Another limitation is the lack of demographic information from the recorded subjects. Albeit we believe that for artefact detection this is not of major influence, it could provide more insight into the differences between the datasets.

Due to the small amount of either clean or noisy samples, depending on the dataset, no deep learning approaches were tested. Future research could investigate the added value of such an approach on a larger dataset.

## 5. Conclusions

We showed that the features that were crafted for contact ECG can be used for artefact detection in ccECG signals. This finding allowed us to successfully construct a classifier for contact ECG and optimise it towards ccECG using transfer learning. Moreover, we showed that the performance of the base classifiers could be significantly improved using only a limited amount of samples.

No added value was obtained by using an active sampling approach based on entropy maximisation compared to random sampling. Therefore, we do not recommend this approach in the future.

In conclusion, transfer learning is able to optimise a modality independent artefact detection model, when only a limited amount of modality-specific samples is available. This results in less labelling and more accurate results.

## Figures and Tables

**Figure 1 sensors-21-00662-f001:**
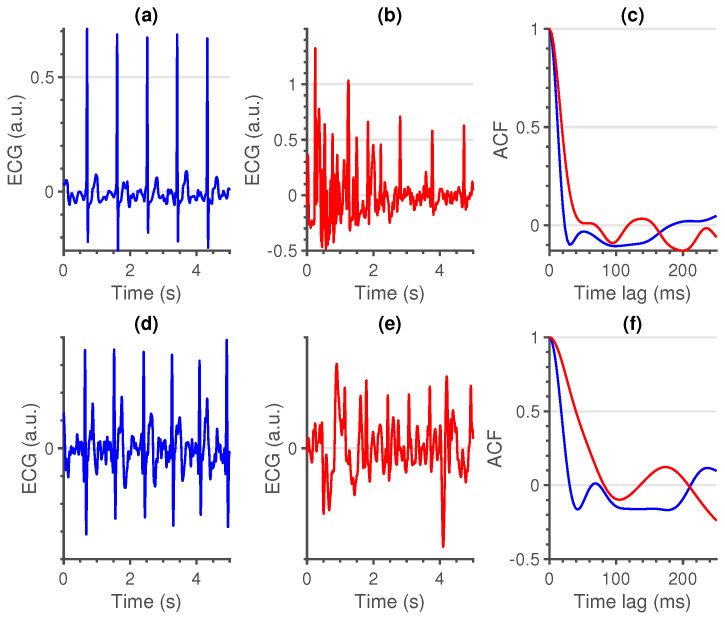
Comparison between a clean (blue) and noisy (red) segment for both contact (**a**,**b**) and non-contact (**d**,**e**) signals. A clear difference in autocorrelation function (ACF) (**c**,**f**) shape can be observed between the two signals.

**Figure 2 sensors-21-00662-f002:**
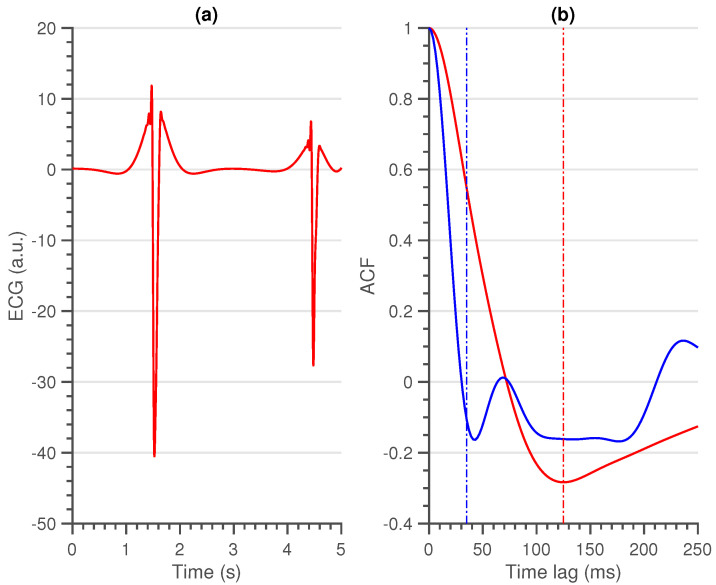
(**a**) An electrocardiogram (ECG) segment that contains a large artefact. (**b**) The respective ACF of the ECG segment (red), together with the ACF of the clean non-contact segment (blue) of Figure 1. The dotted red line in plot indicates the first local minimum and the dotted blue line indicates maximum amplitude at 35 ms.

**Figure 3 sensors-21-00662-f003:**
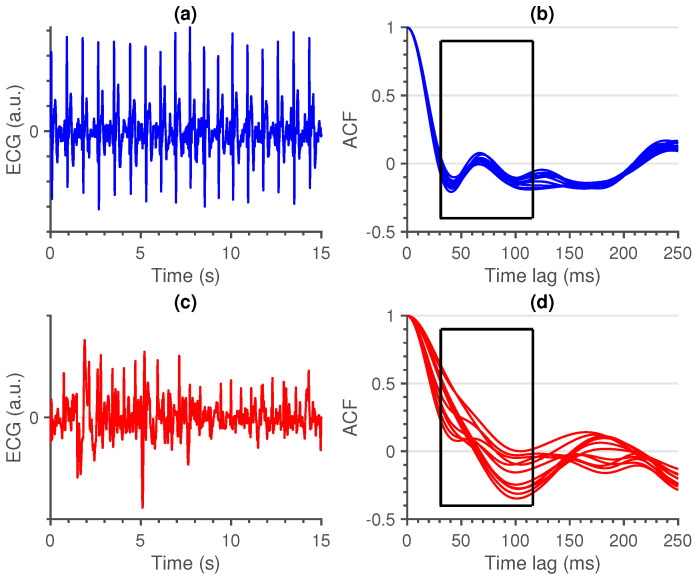
Comparison between a clean (blue) and noisy (red) ECG signal (**a**,**c**), together with the ACF’s of their respective sliding windows (**b**,**d**). The difference between the ACF’s within the search window, as depicted by the black box, is clearly higher for the noisy ECG signal.

**Figure 4 sensors-21-00662-f004:**
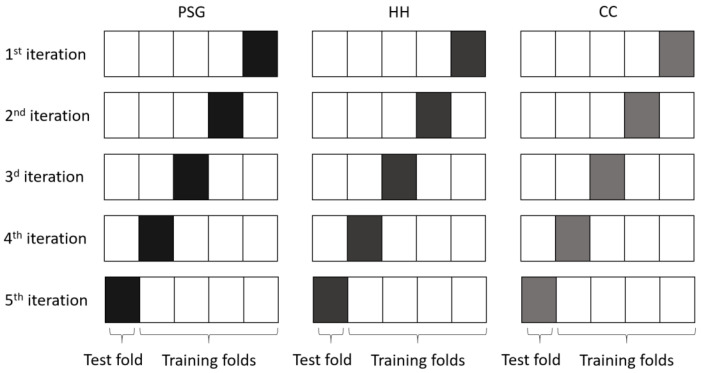
We tested the base classifiers using 5-fold cross-validation. Additionally, each base classifiers was also tested on all folds of the other datasets. This results in a total of 25 performance evaluations.

**Figure 5 sensors-21-00662-f005:**
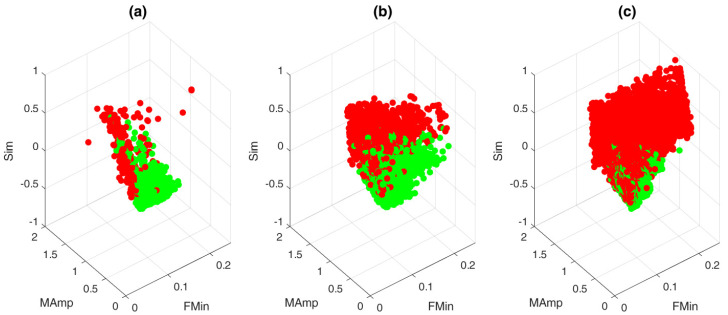
Feature space of the three datasets. Green dots indicate clean samples, red dots indicate noisy samples: (**a**) PSG dataset. (**b**) HH dataset. (**c**) CC dataset.

**Figure 6 sensors-21-00662-f006:**
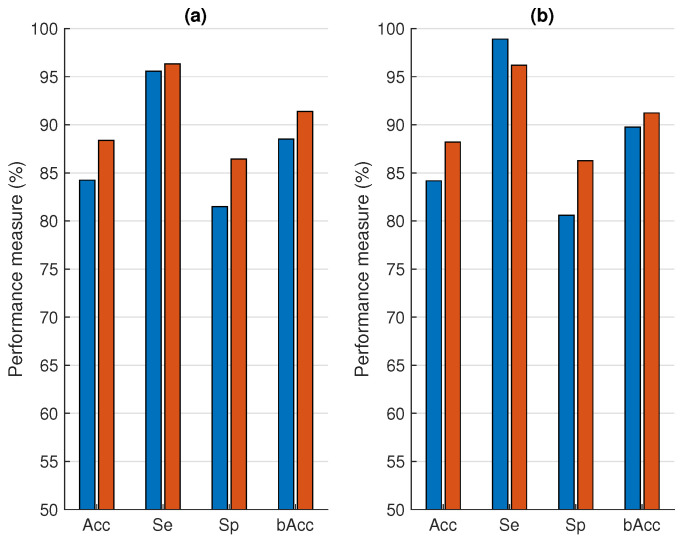
Comparison of the performance on the CC dataset without (blue bars) and with (red bars) transfer learning. (**a**) Trained on the PSG dataset. (**b**) Trained on the HH dataset.

**Figure 7 sensors-21-00662-f007:**
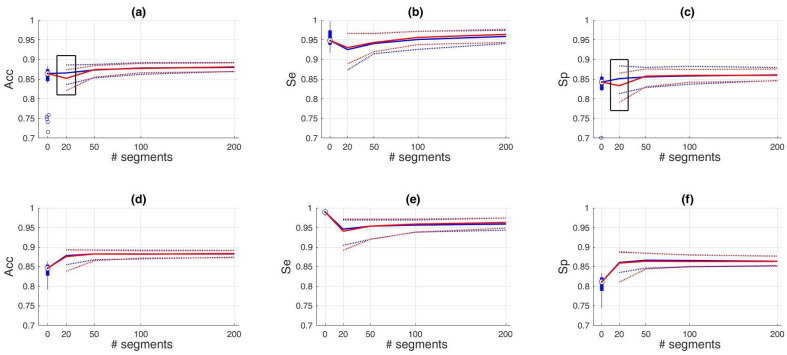
The effect on performance of different subset sizes and the two sampling techniques. The first row (**a**–**c**) corresponds to the performance of the base classifier when trained on the PSG dataset and the second row (**d**–**f**)when trained on the HH dataset. The performance at 0 indicates the performance of the base classifier. As this is evaluated only 25 times instead of the 250 times for the transfer learning performances, we used a blue boxplot. The blue circles below the boxplot indicate outlier values. The blue line that starts in the median value at 0, corresponds to the random sampling procedure and the red line to the fixed size procedure. The dotted lines indicate the interquartile ranges. These provide insight in the variability of the classifier performance. The black boxes indicate a significantly lower performance of the active sampling strategy, compared to random sampling.

**Figure 8 sensors-21-00662-f008:**
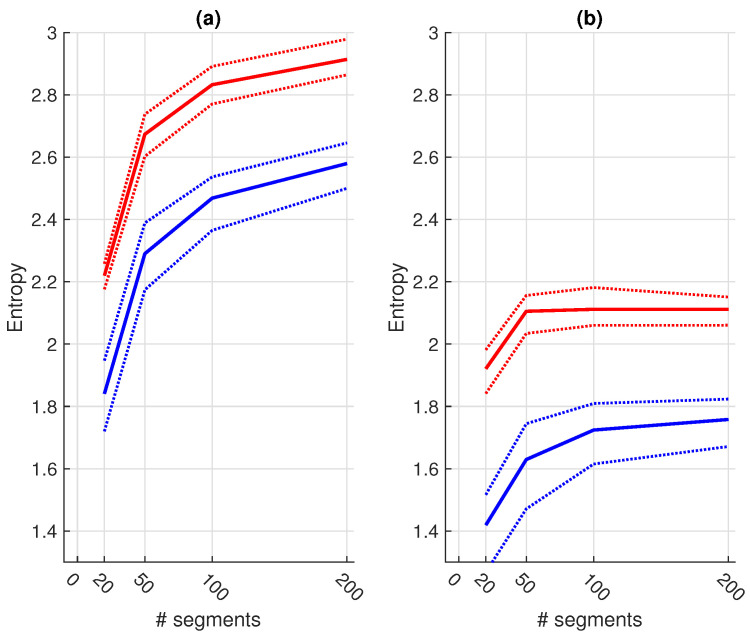
The difference in entropy for the clean (**a**) and noisy (**b**) samples of the CC subsets. The blue line indicates the random sampling and the red line indicates the fixed-size sampling approach. The full line indicates the median values and the dotted lines indicate the interquartile ranges. We can observe that the entropy is consistently higher for the fixed-size sampling.

**Figure 9 sensors-21-00662-f009:**
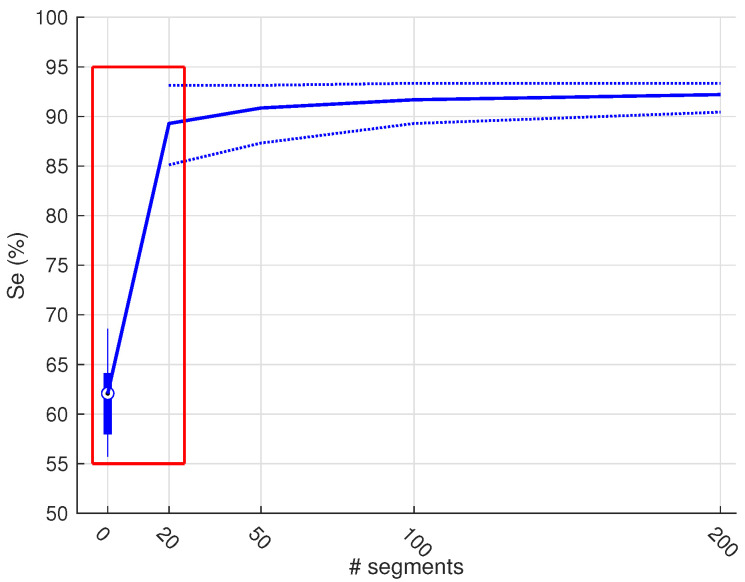
The Se of the CC base classifier when applied on the HH dataset. The full line corresponds to the median values and the dotted lines indicate the interquartile ranges. We used a different graphical representation for the results of the base classifiers at zero, as these results originate from only 25 folds, compared to the 250 folds of the transfer learning approach. A strong increase could already be observed after including only 20 samples.

**Table 1 sensors-21-00662-t001:** Overview of all datasets.

Dataset	Data Source	Sampling Frequency (Hz)	Scenario	# Segments
PSG	University hospitals Leuven	200	During sleep	9132
HH	CinC Challenge of 2017 [20]	300	Unknown	3857
CC	Recordings of system presented in [8]	512	Static car seat	2500
			Bed	2500
			Office chair	1520
			Driving a car	480
	UnoVis database [7]	512	Bed	1000
			Driving a car	1000
			Armchair	1000

**Table 2 sensors-21-00662-t002:** Results obtained for the base classifiers. The columns indicate the training datasets and the rows the test datasets. The cells in grey indicate the intra dataset results when using fivefold cross-validation. The other cells indicate the inter dataset results, averaged over 25 folds. In other words, when the classifier is trained on five training sets and, each time, applied on the five testing sets of the other datasets. The results are shown as mean ± standard deviation.

	PSG			HH			CC		
	Se	Sp	bAcc	Se	Sp	bAcc	Se	Sp	bAcc
**PSG**	99.5±0.1	81.0±11.9	90.3±6.0	99.5±0.12	73.9±5.3	86.7±2.7	91.9±3.0	97.3±2.1	94.6±1.0
**HH**	91.7±3.4	83.4±5.8	87.6±2.6	95.8±1.3	84.4±2.4	90.1±1.1	61.4±3.7	98.3±0.5	79.9±1.9
**CC**	95.6±2.5	81.5±6.9	88.5±2.7	98.9±0.5	80.6±2.2	89.8±1.1	80.0±2.7	94.9±1.2	87.5±1.8

## Data Availability

Not applicable.

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
