# Peer review of "Supervised SVM Transfer Learning for Modality-Specific Artefact Detection in ECG"

_sensors, 2021, doi:10.3390/s21020662_

Round 1

Reviewer 1 Report

Review Report for the Manuscript (1030674): Supervised SVM Transfer Learning for Modality-Specific Artefact Detection in ECG

I have the following comments: 

  1. Authors should add more literature review as the literature section is very limited.
  2. Authors should stress more in a paragraph in the novelty of their work and clear statement about the contribution of the knowledge should be added.
  3. Authors mentioned that some of the datasets obtained from patients. However, they did not mention the ethics approval and the ethics reference number.
  4. Authors should add more information about the datasets such as the gender of the participants, the age ……..etc as this will give better understanding of the results.

Best Wishes 

Reviewer 2 Report

Authors proposed trained ccECG model to improve the artifact detection model performance. Authors selected enough samples for the simulation and showed good performance comparison. In addition, there are no English grammar issues at all. Authors also showed limitation of the proposed research and future work. Therefore, the manuscript can be accepted with the minor comments.

  1. In Figure 8, authors need to indicate the specific important values.
  2. Authors need to improve the Figure 3 quality.
  3. Authors need to increase the size of the Figure 6 labels since they are small.
  4. In Figure 6, authors had better mark the important values.
  5. In Figure 5, authors need to change (A) and (B) to (a) and (b).
  6. In Figure 1, authors need to change (A), (B), and (C) to (a), (b), and (c).
  7. In Conclusion, authors had better express important results.

Reviewer 3 Report

This manuscript proposes a transfer training strategy for capacitively coupled ECG (ccECG) based model pretrained from contact ECG data sets. This kind of transfer learning aims to optimize a modality independent model to detection ccECG artifacts, for the situtation when there are only a limited number of ccECG training samples. The authors conducted some comprehensive evaluations, which prove that the transfer learning is able to achieve good performance to detect ccECG artifacts with a few new ccECG samples. My major and minor comments to this manuscript can be found hereunder.

There are several major comments:

  1. Several indispensable formulas (equations) and the corresponding parameters should be presented and discussed sufficiently in some proper paragraphs. The details are as following:

  1. The formula of auto correlation function (in line 121).
  2. There is no formula or figure for explaining how to compute the Rényi’s quadratic entropy (in 2.4.3 Subset Selection).
  3. Regularization constant C, D in the Equation (1), (2) respectively.
  4. The parameter in Equation (2) depends on N+, N- and γ (as presented in Equation (2) of Reference [10]), but I cannot find any discussion about them in the manuscript.

  1. The strategy of computing Similarity (one of the proposed features, in line 152) is not clear enough. It is highly suggested that the computation should be introduced with a formula, or a figure like Figure 2 in the manuscript.

  1. In Section 2.4.3 (Subset Selection), the authors didn’t illustrate how to select between noisy and clean samples among the new dataset. As shown in Equation (2) of Reference [10], the N+ and N- (the number of clean and noisy samples) in the subset are parameters of the optimization function in SVM. If your experiences found that it is not relevant to evaluation results, please claim in this section to avoid misunderstanding. If it has certain influences on the results, please discuss it.

  1. I have another concern about data selection. As illustrated in Line 251, 500 randomly selected samples were used for transfer learning to update the base classifiers. Does this strategy belong to “Rényi’s quadratic entropy” based selection? The author should make a clear definition between “Rényi’s quadratic entropy” based selection and random selection. Besides, why the evaluation in Figure 6 only takes no more than 200 samples, while the evaluation Figure 5 takes 500 samples instead?

  1. The most confusing point exists in some statements presented in Section 4.2.1: “Albeit this would result ... the classification performance (Line 370)”, “we can conclude that the proposed active sampling strategy does not ... random sampling (Line 374)”. Since the proposed Rényi’s quadratic entropy based selection strategy doesn’t make any progress compared with random selection, why the authors still use this way to select subset data? I believe random selection is much simpler than the proposed way.

  1. For the datasets, the authors should provide the weblinks for those public-available datasets in the section of Reference, including the CinC Challenge 2017 dataset. It is better to provide the other two datasets’ weblinks, especially the datasets of ccECG samples, if available and open to the public.

There are also some minor issues:

  1. There is a typo in Line 44, “A better, and faster, solution”.

  1. Table 2 does not meet the format and the right side exceeds the limit.

  1. I do not understand why the authors use this statement “is evaluated on 25 folds instead of 250 folds” in Figure 6 and Figure 8. The “250 folds” doesn’t exist in the rest of the manuscript. I suppose the author always use 25 folds as the experience condition due to 5 fold cross-validation.

  1. The meaning and role of “interquartile ranges” in Figure 6, 7 and 8 should be clearer. I do not see any description in the corresponding paragraphs.

  1. It is unclear that why some blue circles exist in Figure 6 (A) at the bottom left and Figure 6 (C). The illustration statements of Figure 6 only present the effects of full lines and dotted lines.

Round 2

Reviewer 3 Report

All comments are addressed. Thanks.